# Diagnostic Accuracy of FibroScan and Factors Affecting Measurements

**DOI:** 10.3390/diagnostics10110940

**Published:** 2020-11-12

**Authors:** Satoshi Oeda, Kenichi Tanaka, Ayaka Oshima, Yasue Matsumoto, Eisaburo Sueoka, Hirokazu Takahashi

**Affiliations:** 1Liver Center, Saga University Hospital, 5-1-1 Nabeshima, Saga 849-8501, Japan; takahas2@cc.saga-u.ac.jp; 2Department of Laboratory Medicine, Saga University Hospital, 5-1-1 Nabeshima, Saga 849-8501, Japan; sk1971@cc.saga-u.ac.jp (A.O.); sn6096@cc.saga-u.ac.jp (Y.M.); sueokae@cc.saga-u.ac.jp (E.S.); 3Department of Internal Medicine, Faculty of Medicine, Saga University, 5-1-1 Nabeshima, Saga 849-8501, Japan; kensmvc541112@gmail.com; 4Department of Clinical Laboratory Medicine, Faculty of Medicine, Saga University, 5-1-1 Nabeshima, Saga 849-8501, Japan

**Keywords:** non-alcoholic fatty liver disease, hepatic steatosis, hepatic inflammation, transient elastography, liver stiffness measurement, controlled attenuation parameter

## Abstract

Evaluating liver steatosis and fibrosis is important for patients with non-alcoholic fatty liver disease. Although liver biopsy and pathological assessment is the gold standard for these conditions, this technique has several disadvantages. The evaluation of steatosis and fibrosis using ultrasound B-mode imaging is qualitative and subjective. The liver stiffness measurement (LSM) and controlled attenuation parameter (CAP) determined using FibroScan are the evidence-based non-invasive measures of liver fibrosis and steatosis, respectively. The LSM and CAP measurements are carried out simultaneously, and the median values of more than ten valid measurements are used to quantify liver fibrosis and steatosis. Here, we demonstrate that the reliability of the LSM depends on the interquartile range to median ratio (IQR/Med), but CAP values do not depend on IQR/Med. In addition, the LSM is affected by inflammation, congestion, and cholestasis in addition to fibrosis, while CAP values are affected by the body mass index in addition to steatosis. We also show that the M probe provides higher LSM values but lower CAP values than the XL probe in the same population. However, there was no statistically significant difference between the diagnostic accuracies of the two probes. These findings are important to understand the reliability of FibroScan measurements and the factors influencing measurement values for all patients.

## 1. Introduction

Non-alcoholic fatty liver disease (NAFLD) is pathologically characterized by hepatic steatosis, lobular inflammation, hepatocyte ballooning, and liver fibrosis. According to the fatty liver inhibition of progression (FLIP) algorithm, NAFLD is diagnosed when the steatosis rate exceeds 5% [1]. Liver fibrosis is associated with the prognosis of patients with NAFLD [2,3]. Therefore, assessments of hepatic steatosis and liver fibrosis are important in the daily clinical management of NAFLD. Although liver biopsy and pathology is the gold standard for assessing steatosis and fibrosis, this technique is invasive and expensive and suffers from sampling error and diagnostic variation among observers [4,5,6]. The discordance of one fibrosis stage or more has been reported to be 41% in the evaluation of two tissues taken from different parts of the right lobe of the liver [5]. Kuwashiro et al. have recently shown that the interobserver concordance rates for NAS, steatosis, inflammation, ballooning, fibrosis, and NASH diagnosis were 26.7, 62.7, 51.3, 48.7, 43.3, and 50.7%, respectively [6]. Therefore, a non-invasive and objective method of assessing steatosis and fibrosis is important in NAFLD practice. The efficacy of various non-invasive biomarkers such as circulating biomarkers, scoring systems, and calculating formulas has been reported [7]. Steatosis and fibrosis can also be evaluated using ultrasound B-mode imaging, although the method is limited because it is subjective and semiquantitative [8,9,10]. Yajima et al. have shown that the combination of ultrasonographical findings of liver–kidney contrast, vascular blurring, and deep attenuation enables us to grade fatty change semi-quantitatively [8]. Magnetic resonance elastography (MRE) and ultrasound elastography have been reported as effective methods to compensate for these problems [11,12]. MRE has a higher area under the receiver operating characteristic curve (AUROC) value than one type of ultrasound elastography, FibroScan (Echosens, Paris, France) [12,13], but it is costly to implement. FibroScan is an evidence-based, transient elastography instrument for non-invasive evaluation of liver steatosis and fibrosis [14]. FibroScan can also be used to identify non-alcoholic steatohepatitis with significant activity and fibrosis when its results are combined with aspartate aminotransferase (AST) levels [15,16]. FibroScan is becoming an increasingly important modality in NAFLD practice, as it was recently used to identify patients eligible for NAFLD-related clinical trials.

Recently, a new disease concept called metabolic-associated fatty liver disease (MAFLD) was proposed [17], for which the Asian Pacific Association for the Study of the Liver described the diagnosis and management in its clinical practice guidelines [18]. Thus, fatty liver disease, which is included in chronic liver disease concepts such as NAFLD and MAFLD, is being addressed globally.

FibroScan provides two parameters—the liver stiffness measurement (LSM) and controlled attenuation parameter (CAP)—which are useful for assessing the degree of liver fibrosis and steatosis, respectively. It should be noted that CAP measurement requires special CAP software. FibroScan systems are equipped with two types of probe for adults: an M probe for use in most patients and an XL probe for obese patients. It is important to note that the probes have provided conflicting measurements of both LSM and CAP [19]. In this review, we summarize the diagnostic accuracy and reliability of the measurement values of the LSM and CAP and investigate the factors affecting these measurements. We further compare the M and XL probes for LSM and CAP measurements.

## 2. LSM

### 2.1. Diagnostic Accuracy of LSM in Patients with NAFLD

The LSM is a measure of the speed of the shear wave that is generated by a push pulse as it passes through the liver tissue. The shear wave propagates faster in hard liver tissue than in soft liver tissue. The LSM values range from 1.5 to 75.0 kPa based on this property. Several studies have demonstrated the utility of the LSM for assessing liver fibrosis in patients with various chronic liver diseases including NAFLD. The studies on NAFLD patients are summarized in Table 1 [19,20,21,22,23,24,25,26,27,28,29,30,31,32,33,34]. The AUROCs for detecting fibrosis stages ≥1, ≥2, ≥3, and 4 have been reported as 0.78–0.97, 0.77–0.99, 0.73–1.00, and 0.89–0.997, respectively. Yoneda et al. were the first to report the usefulness of LSM using the M probe in patients with NAFLD [20]. To date, there is not enough evidence to support the usefulness of FibroScan with the XL probe compared with the M probe. Friedrich-Rust et al. reported that AUROCs for detecting fibrosis stages ≥2, ≥3, and 4 using the XL probe were 0.81, 0.84, and 0.95 based on a pilot study [25]. Oeda et al. directly compared the diagnostic accuracies with the M and XL probes and concluded that there was no difference between the AUROCs with the two probes (stage ≥2: 0.777 vs. 0.787, *p* = 0.710; stage ≥3: 0.836 vs. 0.806, *p* = 0.303; stage 4: 0.971 vs. 0.970, *p* = 0.955) [33]. These findings imply that the diagnostic accuracy of the XL probe is comparable to that of the M probe. Recently, a model equipped with an automated probe selection tool (APST) was developed. Eddowes et al. used this model and evaluated the diagnostic performance of the LSM measured using the selected probes; 33% of the subjects were examined using the M probe, and 67% of the patients were examined using the XL probe as indicated by the APST. The AUROCs for detecting fibrosis stages ≥2, ≥3, and 4 were 0.77, 0.80, and 0.89, respectively, suggesting that the diagnostic performance of the LSM obtained using the probes selected by APST is comparable to that reported in previous studies.

### 2.2. Relationship between Measurement Variability and Reliability of the LSM

In the FibroScan measurement protocol, the LSM is obtained based on at least ten valid measurements, and the median value is used to evaluate liver fibrosis. Measurement “failures” are defined as examinations in which ten valid LSMs are not obtained despite ten or more measurements. Studies have shown that, in patients with ten valid LSMs, the interquartile range to median ratio (IQR/Med) affects the reliability of the LSM results [35,36]. Generally, the IQR/Med cutoff value, above which the LSM reliability is low, is 0.3. In clinical practice, however, it is not uncommon for the IQR/Med to be higher than 0.3 if the median value is low; in these cases, the LSM is considered to be unreasonable. Therefore, it is important to evaluate reliability in terms of LSM values. Boursier et al. defined ranges for IQR/Med and Med to evaluate the reliability of examination: IQR/Med ≤ 0.1 with any Med value indicates very reliable measures; IQR/Med > 0.1 and ≤0.3 with any Med value or IQR/Med > 0.3 with Med < 7.1 kPa are considered to indicate reliable measures; and IQR/Med > 0.3 with Med ≥ 7.1 kPa is considered to indicate poor reliability [36]. The success rate of the acquisitions conducted to obtain ten valid measurements does not affect the diagnostic accuracy for liver fibrosis [35,36,37]. Therefore, the consensus is that the success rate does not represent reliability, so the success rate is not displayed on the operation screens of the current FibroScan models.

### 2.3. Factors Affecting the LSM

Several factors affect the LSM in chronic liver disease. We included NAFLD data in this group because few studies on LSM have been conducted exclusively on patients with NAFLD. The LSM is increased by inflammation, venous pressure, cholestasis, and amyloid deposition in the liver [38,39,40,41,42,43,44]. Mueller et al. reported that an increase in bilirubin by 1 mg/dL causes an increase in the LSM by around 1 kPa, an increase in the hepatic venous pressure by 2 cm causes an increase in the LSM by around 1 kPa, and an increase in the AST by 100 U/l causes an increase in the LSM of around 4 kPa [45]. Further, the impact of the AST on the LSM is higher in patients with alcoholic liver disease than in those with hepatitis C [46]. Once these influencing factors are attenuated, the LSM decreases accordingly. Abstaining from alcohol consumption has also been shown to decrease the LSM in patients with alcoholic liver disease [47]. Administering diuretics to patients with high venous pressure also decreases the LSM [41]. Animal experiments indicated that reversing the ligation of the bile duct immediately lowers an elevated LSM [42].

It is recommended that FibroScan be performed after overnight fasting, or at least a few hours after a meal, because food intake also leads to increased LSM values [48,49,50]. Arena et al. measured the LSM after overnight fasting and after standardized liquid meal intake; based on the results, it was suggested that fasting for 120 minutes is required before examination [50]. Mederacke et al. showed that the postprandial LSM is elevated following the intake of a standardized continental breakfast and normalizes to the fasting level 210 min after starting the meal [48].

## 3. CAP

### 3.1. Diagnostic Accuracy of the CAP in Patients with NAFLD

The CAP is a measure of the attenuation of the ultrasound beam. The stronger the liver steatosis is, the more the ultrasound beam passing through the liver tissue is attenuated. CAP values range from 100 to 400 dB/m based on this property [51]. Like the LSM, the CAP is effective for diagnosing liver steatosis in patients with various chronic liver diseases [52]. The studies on patients with NAFLD are summarized in Table 2 [19,31,32,33,34,53,54]. The AUROCs for detecting steatosis scores of ≥1, ≥2, and 3 have been reported as 0.77–0.97, 0.638–0.92, and 0.67–0.83, respectively. Caussy et al. reported that the AUROC was 0.80 for the diagnosis of steatosis of ≥5% and 0.87 for the diagnosis of steatosis ≥ 10% based on the proton density fat fraction measured by magnetic resonance imaging (MRI-PDFF) as a reference standard [55].

### 3.2. Relationship between Measurement Variability and Reliability for the CAP

CAP and LSM measurements are carried out simultaneously, and the median values of ten valid measurements are used to quantify liver steatosis. On the operation screen of the FibroScan, the IQR is displayed, but the IQR/Med is not because it is generally thought that the latter does not affect the measurement reliability [56,57]. Myers et al. showed that, in the detection of steatosis ≥ 10%, there were no significant differences between the AUROCs when IQR/Med ≥ 15% and <15% (*p* = 0.29) and when IQR/Med ≥ 11% and <11% (*p* = 0.63) [56]. Jung et al. showed that the IQR/Med is not an independent predictor of discordance between liver biopsy and CAP value by using multivariate logistic regression analyses [57]. However, there is evidence that the IQR is associated with reliability.

Wong et al. stratified IQR values into three groups (<20, 21–39, and ≥40 dB/m) and determined the AUROCs of these groups (0.86, 0.89, and 0.76, respectively). There was a significant difference between the CAP values in the groups with IQRs of <40 and ≥40 dB/m (0.90 vs. 0.77, respectively, *p* = 0.004) [58], suggesting that variation in the CAP affects the diagnostic accuracy.

### 3.3. Factors Affecting the CAP

As with the LSM, we summarized the factors affecting the CAP in chronic liver disease and included NAFLD data as well. The measured CAP values can be affected by various factors in addition to liver steatosis. Studies that examined influencing factors using multivariate analyses are shown in Table 3 [53,56,57,59,60,61,62,63]. The data were adjusted for liver steatosis in all of the studies. Body mass index (BMI) was reported to be independently associated with CAP in six studies. Triglyceride was reported to affect the CAP in one study and not affect it in four studies. Hence, it was concluded that triglyceride does not affect the CAP. Age, sex, liver fibrosis, total cholesterol, and fasting glucose level were not found to be factors that affected the CAP.

Table 4 shows whether factors affecting the LSM or the CAP also affect the CAP or the LSM, respectively. Inflammation and liver function, which affected the LSM, were reported not to affect the CAP [53,56,57,59,60,61,62,63]. There was insufficient evidence that venous pressure, cholestasis, amyloidosis, or food intake affected the CAP. One report found that food intake decreased the CAP, and another found that it had no effect on it [64,65]. Liver steatosis and BMI, which affected the CAP, were reported not to affect the LSM after adjusting for the stage of fibrosis [23].

## 4. Probe Comparison and Selection

FibroScan is equipped with two types of probes for adults: an M probe for use on the majority of patients and an XL probe designed for obese patients. In addition, an S probe is generally used on pediatric patients. The XL probe can take measurements at greater depths (35–75 mm) than the M probe (25–65 mm) [25]. The diameter of the transducer of the XL probe is larger (10 mm) than that of the M probe (7 mm). The center frequency of the ultrasound waves is 2.5 MHz for the XL probe and 3.5 MHz for the M probe. In addition, it is recommended to use the M probe on any patient with a skin-capsular distance (SCD) of ≤25 mm and the XL probe on any patient with an SCD of >25 mm. However, as the automated probe selection tool (APST) is embedded in the software of the late FibroScan 502 model and later models, the measurement of SCD using ultrasound B-mode imaging is not required when using these models.

Both the LSM and CAP are designed to have identical values from the M and XL probes. The shear wave frequency of the LSM is 50 Hz for both probes. Although the center frequency differs between the M probe (3.5 MHz) and the XL probe (2.5 MHz), the CAP values captured using the XL probe are adjusted to 3.5 MHz [66]. Therefore, the ultrasound attenuation with the probes should be the same in the same region in the same patient, while the resulting CAP values could differ because the probes have different regions of interest.

Generally, the LSM obtained using the M probe is higher than that obtained using the XL probe in the same population [19,28,33,67,68,69]. One study showed that LSMs measured using the XL probe were 1.7 ± 2.3 kPa lower than those measured by the M probe by pairwise examination [28]. Another study reported that the mean difference between the LSM measurements was 2.3 kPa (the median difference was 1.4 kPa) [67]. A third study reported that the median difference between the XL and M probe LSM measurements was 2.6 kPa [68]. Wong et al. estimated the LSM values that would be captured with the M probe from results obtained with the XL probe (LSM-XL) by using linear regression analysis, yielding the following equations: 1.110 × LSM-XL + 0.954 for patients with BMIs 25.0–30.0 kg/m^2^ and 1.204 × LSM-XL + 0.931 for patients with BMIs > 30.0 kg/m^2^ [68]. Previous studies showed that the XL probe returns higher CAP values than the M probe [19,33,70]. However, regardless of the probe used, the cutoff values, sensitivities, specificities, positive predictive values, and negative predictive values were comparable for the diagnosis of steatosis grades ≥ S1, ≥ 2, and 3 [71]. Unlike LSM, there is insufficient evidence comparing the XL probe with the M probe for CAP measurements, and it remains unclear how much the higher CAP values obtained by XL probe are compared with those obtained by the M probe. Oeda et al. recently reported probe-specific cutoff values for the CAP and LSM and showed that there was no significant difference between the AUROCs with the two probes using these cutoff values [33].

## 5. Conclusions

In patients with NAFLD, it is important to evaluate fibrosis and steatosis because fibrosis is associated with clinical prognosis, and steatosis is a criterion for NAFLD diagnosis. Various types of tests are available for the surveillance of NAFLD patients [72]. Among them, the LSM and CAP obtained by FibroScan are useful and cost-effective measures for the diagnosis of liver fibrosis and steatosis. However, users should bear in mind the factors that can influence measurements besides fibrosis and steatosis: the LSM is affected by inflammation, venous pressure, cholestasis, amyloid deposition, and food intake, and the CAP is affected by the BMI. Moreover, it is necessary to evaluate the reliability of the obtained LSM values based on the IQR/Med, but the IQR/Med is not associated with the reliability of CAP values. In addition, it is important for FibroScan users to understand the reliability of measurement values and factors influencing the measurement values. Especially in cases when there is a discrepancy between the FibroScan results and clinical data, such as liver biopsy, biomarker measures, and observations with other imaging modalities, FibroScan results should be interpreted carefully as a possible indicator of liver fibrosis and steatosis in clinical application. In the future, it would be desirable to study the LSM and the CAP values corrected for factors affecting their measurement values.

## Figures and Tables

**Table 1 diagnostics-10-00940-t001:** Diagnostic accuracy for liver fibrosis in patients with NAFLD.

Author	Year	N	Probe	AUROC (Prevalence (%))			
				Stage ≥1	Stage ≥2	Stage ≥3	Stage 4
Yoneda et al. [20]	2007	67	M	0.881 (78)	0.876 (49)	0.914 (24)	0.997 (7)
Yoneda et al. [21]	2008	97	M	0.927 (81)	0.865 (53)	0.904 (28)	0.991 (9)
Nobili et al. [22]	2008	50	M	0.97 (78)	0.99 (24)	1.00 (10)	-
Wong et al. [23]	2010	246	M	-	0.84 (41)	0.93 (23)	0.95 (10)
Lupsor et al. [24]	2010	72	M	0.879 (65)	0.789 (25)	0.978 (7)	-
Friedrich-Rust et al. [25]	2010	50	M	-	0.79 (30)	0.75 (24)	0.91 (6)
		50	XL	-	0.81 (30)	0.84 (24)	0.95 (6)
Petta et al. [26]	2011	146	M	-	0.794 (47)	0.870 (23)	-
Friedrich-Rust et al. [27]	2012	37	M	-	0.80 (n.d.)	0.73 (n.d.)	0.93 (n.d.)
		43	XL	-	0.82 (n.d.)	0.84 (n.d.)	0.93 (n.d.)
Wong et al. [28]	2012	156	M	-	0.83 (42)	0.87 (27)	0.89 (10)
		184	XL	-	0.80 (45)	0.85 (29)	0.91 (13)
Kumar et al. [29]	2013	205	M	0.82 (84)	0.85 (68)	0.94 (55)	0.96 (46)
Pathik et al. [30]	2015	110	M	-	-	0.91 (35)	-
Imajo et al. [31]	2016	127	M	0.78 (n.d.)	0.82 (n.d.)	0.88 (n.d.)	0.92 (n.d.)
Chan et al. [19]	2017	57	M	0.88 (60)	0.95 (23)	0.97 (14)	0.97 (5)
		57	XL	0.87 (60)	0.90 (23)	0.95 (14)	0.98 (5)
Eddowes et al. [32]	2019	373	M/XL	-	0.77 (60)	0.80 (38)	0.89 (9)
Oeda et al. [33]	2020	96	M	-	0.777 (52)	0.836 (27)	0.971 (5)
		96	XL	-	0.787 (52)	0.806 (27)	0.970 (5)
Cardoso et al. [34]	2020	81	M	-	0.82 (19)	-	-
		81	XL	-	0.80 (19)	-	-

Abbreviations: AUROC, area under the receiver operating characteristic curve; NAFLD, non-alcoholic fatty liver disease; N, the number of patients; n.d., not described.

**Table 2 diagnostics-10-00940-t002:** Diagnostic accuracy for liver steatosis in patients with NAFLD.

Author	Year	N	Probe	AUROC (Prevalence (%))		
				Score ≥1	Score ≥2	Score 3
Chan et al. [53]	2014	101	M	0.97 (87)	0.86 (64)	0.75 (14)
Imajo et al. [31]	2016	127	M	0.88 (n.d.)	0.73 (n.d.)	0.70 (n.d.)
Chan et al. [19]	2017	57	M	0.94 (98)	0.80 (75)	0.69 (26)
		57	XL	0.97 (98)	0.81 (75)	0.67 (26)
Eddowes et al. [32]	2019	380	M/XL	0.87 (80)	0.77 (64)	0.70 (36)
Darweesh et al. [54]	2019	60	M	-	0.77 (63)	0.92 (22)
Oeda at al [33]	2020	100	M	-	0.638 (44)	0.687 (19)
		100	XL	-	0.680 (44)	0.713 (19)
Cardoso et al. [34]	2020	81	M	-	0.75 (73)	0.83 (23)
		81	XL	-	0.76 (73)	0.82 (23)

Abbreviations: NAFLD, non-alcoholic fatty liver disease; N, the number of patients; n.d., not described.

**Table 3 diagnostics-10-00940-t003:** Factors affecting CAP measurements.

Etiology	[53] *	[56] *	[57] *	[59] *	[60] *	[61] *	[62] *	[63] *
	NAFLD	NAFLD Others	NAFLD HBV HCV Others	HCV	NAFLD HBV HCV	NAFLD HBV HCV Others	NAFLD HBV	HBV
Age				×	×			×
Sex				×	×			×
Body mass index	○		○		○	○	○	○
Liver steatosis	○	○	○	○	○	○	○	○
Inflammation		×	×	×	×	×		×
Liver fibrosis		×	×	×	×	×		
AST	×				×	×	△	
ALT	×		×		×	×	△	×
Total cholesterol	×		×		×	×	△	×
Triglyceride	○				×	×	△	×
Fasting glucose	×					×		

* Reference number; ○: significant and independent factor; △: factor excluded by the stepwise regression analysis; ×: not a significant factor. Abbreviations: CAP, controlled attenuation parameter; AST, aspartate aminotransferase; ALT, alanine aminotransferase; NAFLD, non-alcoholic fatty liver disease; HBV, hepatitis B virus; HCV, hepatitis C virus.

**Table 4 diagnostics-10-00940-t004:** Associations between factors affecting the LSM or CAP *.

Factors	LSM	CAP
Liver fibrosis	↑	→
Inflammation	↑	→
Venous pressure	↑	→
Cholestasis	↑	→
Amyloidosis	↑	→
Food intake	↑	→
Liver steatosis	→	↑
Body mass index	→	↑

↑: increased FibroScan measurements; →: did not affect FibroScan measurements, →: not enough evidence. Abbreviations: CAP, controlled attenuation parameter; LSM, liver stiffness measurement. * As described in Section 2.3 and Section 3.3 and FibroScan measurements.

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
