# Peer review of "Diagnostic Accuracy of FibroScan and Factors Affecting Measurements"

_diagnostics, 2020, doi:10.3390/diagnostics10110940_

Round 1
Reviewer 1 Report
Oeda et al have presented very clearly and in detail in this review the diagnostic accuracy and reliability of the measurement values LSM and CAP.
Moreover, they further investigated the factors affecting the measurements and compared the M and XL probes for LSM and CAP measurements.
The review deals with an increasingly important issue in the field of clinical hepatology.
Epidemiologically, non-alcoholic fatty liver disease is an increasing clinical manifestation.
It will increase in the future or clinically critical liver dysfunctions will increasingly result from it. Early and reliable diagnosis is a very important factor in preventing irreversible liver tissue destruction and thus liver cirrhosis and liver carcinomas.
Nevertheless, I have 2 comments which should be taken into account and thus be discussed in more detail in the review:
1. in the subdivision of the different degrees of fibrosis, the subgroups with greater or equal 1, 2, 3 are listed as for example in table 1.
It would be more relevant to show exactly how many patients were exactly 1 in the subgroup greater than or equal to 1 and etc.
At this point an additional column would have to be inserted and the proportion of exactly 1, exactly 2 and exactly 3 would have to be shown and statistically weighted. Otherwise I see this as an immense bias, which makes the clinical interpretation of these data more difficult.
As stated in 2.3, for example, the increase in inflammation in the liver influences LSM. In 3.3. again, hepatic inflammation is not seen as an influencing or disturbance variable for CAP.
Please discuss this aspect in more detail in the discussion and point 3, respectively, and also provide an overview in tabular form of a) which factors have what influence on CAP and LSM and b) why methodologically or pathophysiologically these differences can or cannot be expected.
Author Response
Thank you for your good advices. I attached reply file.

Reviewer 2 Report
The paper is a very thorough review of the literature. It covers all sensitive topics in relation with transient elastography. The conclusion are important for clinical practice.
I would suggest adding future research points in this field.
Author Response

(The authors gave the same response as above.)

Reviewer 3 Report
The manuscript by Satoshi Oeda et al. deals with the use of Fibroscan and factors affecting measurements.
Comments:
1) The title is not concordant with the topic addressed in the manuscript. Introduction and abstract deal with Metabolic fatty liver disease therefore I suggest to change the title.
2) line 43: "Steatosis and fibrosis can also be evaluated using ultrasound B-mode imaging". I suggest to use "ultrasound-based techniques"
3) line 44: "this method is limited because it is qualitative and subjective". This sentence can be used only for steatosis. Indeed, ultrasound-based techniques for the quantification of liver fibrosis give us a number and the evaluation is therefore quantitative. Please change this sentence.
4) line 49:" for non-invasive clinical evaluation of liver steatosis and fibrosis". Fibroscan is a technique used for the evaluation of liver fibrosis. The machine can be implemented with CAP software but in the absence of this specific software Fibroscan gives us only liver stiffness values and data on liver fibrosis. For the sake of accuracy, please change this sentence. Moreover delete the word "clinical".
5) Line 49-51: this sentence is out of context. Please delete it or keep it enhancing the concept dealing with different ways to strengthen the results obtained by Fibroscan.
6) line 52-54: see point 4
7) line 55-57: the aim of the manuscript is not clear. The factors affecting the measurements are referred only to liver steatosis?
8) I suggest to use the term MAFLD. It has replaced the term NAFLD (see Guidelines EASL 2020: Eslam M, Newsome PN, Sarin SK, et al. A new definition for metabolic dysfunction-associated fatty liver disease: An international expert consensus statement. J Hepatol. 2020;73(1):202-209. doi:10.1016/j.jhep.2020.03.039
9) In my opinion, the introduction is too short and does not introduce the topic.It should be deeply changed.
10) line 61-63: These sentences should be included in the introduction section.
11) line 86-87: please improve this sentence
12) the paragraph "Factors affecting the LSM" are too generic. Why do the authors report data on NAFLD in the previous parts whereas in this paragraph they address general data? This paragraph is also poor and is not concordant with introduction and previous paragraphs.
13) "Probe comparison and selection": this paragraph includes data that should be discussed before. I suggest to use these concepts in the introduction section.
14) The "CAP" section is short, poor and out of contest.These data could be the topic for a new and specific manuscript.
Author Response

(The authors gave the same response as above.)
